# Revisiting Design Choices in Offline Model Based Reinforcement Learning

## Abstract

Offline reinforcement learning enables agents to leverage large pre-collected datasets of environment transitions to learn control policies circumventing the need for potentially expensive or unsafe online data collection. In recent times there has been significant progress in offline RL, with the dominant approach becoming methods which leverage a learned dynamics model. This typically involves constructing a probabilistic model, and using the model uncertainty to penalize rewards where there is insufficient data, solving for a *pessimistic* MDP that lower bounds the true MDP. Recent work, however, exhibits a breakdown between theory and practice, whereby pessimistic return ought to be bounded by the *total variation distance* of the model from the true dynamics, but is instead implemented through a penalty based on estimated *model uncertainty*. This has spawned a variety of uncertainty heuristics, with little to no comparison between differing approaches. In this paper, we compare these heuristics, and design novel protocols to investigate their interaction with other hyperparameters such as the number of models, or imaginary rollout horizon. Using these insights, we show that selecting these key hyperparameters using Bayesian Optimization produces optimal configurations that are vastly different to those currently used in existing hand-tuned state-of-the-art methods, often resulting in drastically stronger performance.

## 1 Introduction

In offline (or batch) reinforcement learning (RL) [13, 26], the goal is to learn policies that perform well in an environment given a fixed data set of pre-collected experiences. This could have vast implications for using RL in real-world settings, as agents can make use of ever increasing amounts of data without the need for an accurate simulator, while also avoiding expensive and potentially even unsafe exploration in the environment.

Model-based reinforcement learning (MBRL) has recently shown promise in this paradigm, obtaining state-of-the-art performance on offline RL benchmarks [21, 48], improving upon powerful model-free approaches (i.e., [23]). MBRL works by training a dynamics model from the offline data, then optimizing a policy using imaginary rollouts from the model. This allows the agent to learn from on-policy experience, as the model is agnostic to the policy used to generate data. Furthermore, recent work has demonstrated the utility of world models *beyond* maximizing return, such as generalizing to unseen environments [4], transferring to new tasks in the same environment [49], and learning with safety constraints [2]. Therefore, the case for MBRL in offline RL is clear: not only does it represent state-of-the-art in terms of performance, but it also provides the opportunity to maximize the signal in the offline data to generalize onto tasks beyond those encoded by the behavior policy.

However, a common failure mode of MBRL is when the policy can exploit the model in parts of the state-action space where the model is inaccurate. Thus, naïve application of MBRL to offline data can

result in sub-optimal performance. To prevent this, concurrent recent works [49, 21] have approached the problem by training a policy in a *pessimistic* MDP (P-MDP). The P-MDP lower bounds the true MDP, and discourages the policy from regions where there is large discrepancy between the true and learned dynamics; this often provides a theoretical guarantee of improvement over simply cloning the behavior policy. This is made practically possible by adding a penalty proportional to the uncertainty in the dynamics model. However, while these recent successes are similar in principle, in practice they differ in a series of design choices. First and foremost, they make use of different heuristics to measure model uncertainty, in some cases deviating from simpler metrics which are more consistent with the theory. Indeed, these decisions are justified by superior performance, given a limited amount of hyperparameter tuning or analysis.

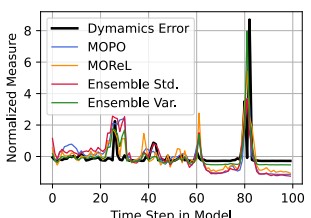

Figure 1: How penalty and true error vary over a model rollout

In this paper we conduct a rigorous investigation into a series of these design choices. We begin focusing on the choice of uncertainty metric, comparing both recent state-of-the-art offline approaches [21, 49, 41] with additional metrics used in the online setting [3, 37, 9]. We also explore the interaction with a series of other hyperparameters, such as the number of models and imaginary rollout length. Interestingly, the relationship between these variables and the model uncertainty varies significantly depending on the choice of metric. Furthermore, we compare these uncertainty heuristics under new evaluation protocols that, for the first time, capture the specific covariate shift induced by model-based RL. This allows us to assess calibration to model exploitation in MBRL, observe that some existing penalties are surprisingly successful at capturing the errors in predicted dynamics, as seen in Fig. 1. Finally, using the insights gained from this section, we test the capability of existing methods given an optimal choice over all variables, modeled jointly using a powerful Bayesian Optimization algorithm [46]. We find that a simple and intuitive uncertainty measure can provide state-of-the-art results in continuous control benchmarks when properly tuned, and the chosen hyperparameters align with our analysis.

We believe this work will contain a variety of interesting insights for researchers and practitioners in offline RL. Below we highlight some of the main findings:

- **Longer horizon rollouts with larger penalities can improve existing methods.** We see that conducting significantly longer rollouts inside the model, coupled with larger uncertainty penalities, typically improves performance.

- **Penalties that are more closely aligned with the theory achieve better correlation with OOD measures.** The deep ensembles approach of [25] often outperforms the penalty from MOPO [49] and MOReL [21]. We observe that the ensemble standard deviation is statistically strikingly similar to the MOReL penalty, but has improved correlation and scaling behavior.

- **Uncertainty is more correlated with *dynamics error* than *distribution shift*.** We find that successful penalties measure the discrepancy in dynamics, and can in fact assign high certainty to regions far away from the offline data.

## 2   Related Work

Two recent works concurrently demonstrated the effectiveness of model based reinforcement learning (MBRL) in the offline setting. *MOPO* [49] follows MBPO [19] but trains inside a conservative MDP which penalizes the reward based on the maximum aleatoric uncertainty over the ensemble members. *MOReL* achieves even stronger performance, penalizing the rewards by a penalty based on the maximum pair-wise difference in ensemble member predictions. For pixel-based tasks, *LOMPO* [41] also proposed a novel penalty, using the variance of ensemble log-likelihoods. Outside of the offline setting, probabilistic dynamics models leveraging uncertainty have underpinned a series of successes [8, 35, 24, 6, 37]. Uncertainty can also be measured in MBRL without the use of neural networks [10], although these methods tend to be harder to scale and thus lack widespread use.

Effective hyperparameter selection in RL has been shown to be crucial to the success of commonly used algorithms [1, 12]. This becomes even more challenging in MBRL with additional hyperparameters for the dynamics model and model architecture needing to be selected. Recent work has shown that carefully optimizing these hyperparameters for online MBRL can significantly improve performance, with the tuned agent breaking the MuJoCo simulator [50]. In contrast, we focus on the

offline setting, and investigate parameters specifically related to uncertainty estimation. Previous work studied the impact of hyperparameters in offline RL [36], finding offline RL algorithms to be brittle to hyperparameter choices. However, unlike our work they only consider model-free approaches, whereas we specifically investigate *model-based* offline algorithms.

Our work also relates to the rich literature on *deep ensembles* [25], which train multiple deep neural networks with different initializations and different dataset orderings, and generally outperform variational Bayesian methods [27, 5]. Achieving effective uncertainty calibration with neural networks is notoriously difficult [16, 22, 28], and furthermore we require good calibration in the face of co-variate shift [34] as the policy we learn in the model will likely deviate from the behavior policy that generated the offline data. Indeed, recent work has highlighted this issue in offline RL [23, 48] and has reported superior performance despite eschewing model uncertainty entirely. However, it is unclear if this performance improvement is due to poor uncertainty calibration, implementation details, or a fundamental limitation of the pessimistic-MDP formulation.

## 3 Background

All of the methods we investigate in this paper model the environment as a Markov Decision Process (MDP), defined as a tuple $M = (\mathcal{S}, \mathcal{A}, P, R, \rho_0, \gamma)$, where $\mathcal{S}$ and $\mathcal{A}$ denote the state and action spaces respectively, $P(s'|s, a)$ the transition dynamics, $R(s, a)$ the reward function, $\rho_0$ the initial state distribution, and $\gamma \in (0, 1)$ the discount factor. The goal is to optimize a policy $\pi(a|s)$ that maximizes the expected discounted return $\mathbb{E}_{\pi, P, \rho_0} \left[ \sum_{t=0}^{\infty} \gamma^t R(s_t, a_t) \right]$.

In *offline RL*, the policy is not deployed in the environment until test time. Instead, the algorithm only has access to a static dataset $\mathcal{D}_{env} = \{(s, a, r, s')\}$, collected by one or more behavioral policies $\pi_b$. Following the notation in [49] we refer to the distribution from which $\mathcal{D}_{env}$ was sampled as the *behavioral distribution*. The most prominent offline MBRL methods all train an ensemble of $N$ probabilistic dynamics models [32]. These usually learn to predict both the next state $s'$ and reward $r$ from a state-action pair, and are trained on $\mathcal{D}_{env}$ using supervised learning. Concretely, each of the $N$ models output a Gaussian $\widehat{P}_\phi^i(s_{t+1}, r_t | s_t, a_t) = \mathcal{N}(\mu_\phi^i(s_t, a_t), \Sigma_\phi^i(s_t, a_t))$ parameterized by $\phi$. The resulting learned dynamics model $\widehat{P}$ and reward model $\widehat{R}$ define a *model MDP* $\widehat{M} = (\mathcal{S}, \mathcal{A}, \widehat{P}, \widehat{R}, \rho_0, \gamma)$. To train the policy, we use $k$-step rollouts inside $\widehat{M}$ to generate trajectories [43].

To prevent policy exploitation in a model, a pessimistic MDP (P-MDP) is constructed by lower bounding the true-expected return using some error between the true and estimated models. For instance, in [49] the authors show that a lower bound on the return can be established by penalizing the reward by a measure that corresponds to estimated model error:

$$\eta_M(\pi) \leq \mathbb{E}_{(s,a) \sim \rho_{\widehat{T}}^\pi} \left[ r(s, a) - \gamma | G_{\widehat{M}}^\pi(s, a) | \right] \tag{1}$$

Several potential choices for $|G_{\widehat{M}}^\pi(s, a)|$ are proposed, including an upper bound based on the total variation distance between the learned and true dynamics. However, for their practical algorithm the authors elect to use an alternative, based on impressive empirical results. Concurrent to MOPO, MOReL [21] constructs a P-MDP by augmenting a standard MDP with a negative valued absorbing state that is transitioned to when total variation distance between true and learned dynamics is exceeded. They show that a policy learned in the P-MDP exceeds simple behavior cloning. Whilst dynamics-based total variation distance has desirable theoretical properties, the practical algorithm relies on a heuristic to approximate this quantity. Next, we investigate the penalties used in these works, as well as other under-used candidates, and explore their effectiveness.

## 4 Uncertainty Penalty

As we have discussed, the key idea underpinning recent success in offline MBRL is the introduction of a conservative MDP, penalized by some uncertainty penalty. The theory dictates this should be some distance measure between the true and predicted dynamics. Of course, this cannot be truly estimated without access to an oracle, so instead a proxy for this quantity is constructed instead. In this paper, we compare the following uncertainty heuristics, from recent works in both offline and online MBRL:

**MOPO [49]:** $\max_{i=1,\ldots,N}||\Sigma_\phi^i(s,a)||_{\mathrm{F}}$, which corresponds to the maximum aleatoric error, computed over the variance heads of the model ensemble.

**MOReL [21]:** $\max_{i,j}||\mu_\phi^i(s,a) - \mu_\phi^j(s,a)||_2$, which corresponds to the pairwise maximum difference of the ensemble predictions.

**LOMPO [41]:** $\mathrm{Var}(\{\log \widehat{P}_\phi^i(s'|s,a), i=1,\ldots,N\})$, where $s'$ is a next state sampled from a single ensemble member. We evaluate its log-likelihood under each ensemble member and take the variance.

**M2AC [37]:** $D_{\mathrm{KL}}[\widehat{P}_{\phi_i}(\cdot|s,a)||\widehat{P}_{\phi_{-n}}(\cdot|s,a)]$, which corresponds to the KL divergence between the Gaussian parameterized by the randomly selected ensemble member we generate the next state from, and the aggregated Gaussian of the remaining ensemble members.

**Ensemble Standard Deviation/Variance [25]:** $\Sigma^*(s,a) = \frac{1}{N}\sum_i^N ((\Sigma_\phi^i(s,a))^2 + (\mu_\phi^i(s,a))^2) - (\mu^*(s,a))^2$ where $\mu^*$ is the mean of the means ($\mu^*(s,a) = \frac{1}{N}\sum_i^N \mu_\phi^i(s,a)$). This corresponds to a combination of epistemic and aleatoric model uncertainty. This is surprisingly under-utilized in offline MBRL, and is arguably the most principled uncertainty penalty. We choose to evaluate both standard deviation and variance as this will provide intuition about the importance of penalty distribution *shape*.

Each of these penalties can be computed using the output from an ensemble of probabilistic dynamics models [25, 8], thus, we are able to compare them in a controlled manner.

### 4.1 How Do These Perform on Fixed Offline Datasets?

We begin by assessing how well uncertainty penalties correlate with next state prediction error. This is crucial in order to correctly penalize the policy from visiting parts of the state-action space where the model is inaccurate, and therefore exploitable. We use the datasets from D4RL [14], train models on each dataset, then evaluate them on *other datasets* from the same environment, but collected under *different* policies. This is important as we may change the task we train on in the model (such as the Ant-direction experiment in [49]), so require good calibration on *unseen* data. As a result, we call these our 'Transfer' experiments. We compare the penalty and MSE for a variety of settings in the Appendix (see: Section A.2), with a snapshot in Fig. 2. We measure Spearman rank ($\rho$) and Pearson bivariate ($r$) correlations, and discuss this in App. A.1. Full details of all experiments and hyperparameters are given in App. G.

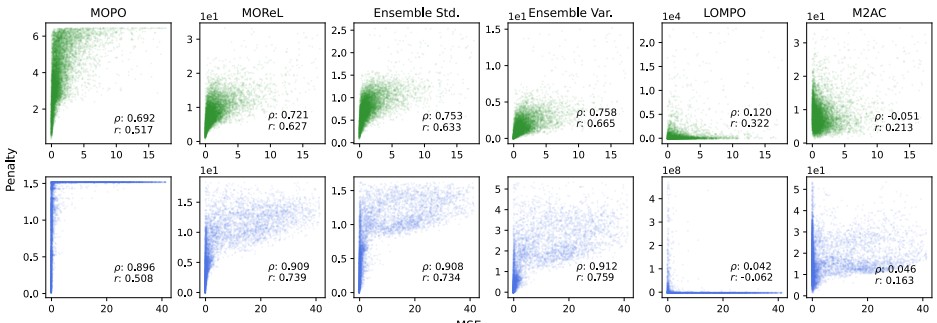

Figure 2: Scatter Plots showing models trained on D4RL Medium being tested on data from Random. Green = HalfCheetah, Blue = Hopper.

Before we begin analyzing these results in detail, we now introduce a novel approach to assessing our penalties under the OOD data induced by model exploitation by a policy.

### 4.2 How Do These Perform During an Imaginary Rollout?

We now design an experiment aimed at capturing the OOD data *generated by the actual offline MBRL process*, which we call our 'Ground Truth' experiments. First, we train a set of policies *without* a penalty inside the model. We then measure the difference between the return predicted by the model over a rollout, and the true return in the real environment. We define a policy to be 'exploitative' if the model significantly *over-estimates* the return compared to the true return. It is vital that we train exploitative policies as these precisely induce the extrapolation errors which cause MBRL methods to fail in the offline setting. It is therefore important that the penalty is able to accurately determine when

the model is being exploited in this way. We use a subset of the most exploitative policies to generate trajectories in the model, and record the uncertainty predicted by each penalty at each time step. To generate the ground truth data, we then 'replay' these trajectories in the true environment, loading the state and action taken in the model into the environment, and record the 'true' next state according to the MuJoCo simulator [44]. The 'Ground Truth' is therefore the MSE between the predicted next state and actual next state. Additional details are provided in App. D along with plots in App. A.2. Table 1 summarizes the results from both the 'Transfer' and 'Ground Truth' experiments.

Table 1: Statistics of all experiments averaged over different test settings.

| | Transfer | | | | Ground Truth | | | |
| | HalfCheetah | | Hopper | | HalfCheetah | | Hopper | |
| Penalty | $\rho$ | r | $\rho$ | r | $\rho$ | r | $\rho$ | r |
|---|---|---|---|---|---|---|---|---|
| MOPO | 0.780 | 0.545 | 0.710 | 0.411 | 0.581 | 0.419 | 0.732 | 0.484 |
| MOReL | 0.789 | 0.624 | 0.772 | 0.571 | 0.581 | 0.518 | 0.750 | 0.546 |
| Ensemble Std. | 0.820 | 0.644 | 0.789 | 0.556 | 0.608 | 0.521 | 0.789 | 0.545 |
| Ensemble Var. | 0.821 | 0.671 | 0.786 | 0.589 | 0.604 | 0.493 | 0.767 | 0.545 |
| LOMPO | 0.126 | 0.141 | 0.361 | 0.122 | 0.035 | 0.067 | 0.496 | 0.161 |
| M2AC | 0.029 | 0.107 | 0.111 | 0.082 | -0.019 | 0.062 | 0.220 | 0.095 |

We immediately notice that the LOMPO and M2AC penalties have very weak correlation with MSE for the examples in Fig. 2. We believe this is the case because LOMPO relies on likelihood statistics, which are notoriously sensitive, and has been designed for use in scenarios involving 'well-behaved' latent dynamics that are KL-regularized to a spherical Gaussian. Regarding M2AC, we note that this penalty was designed for the online setting with significantly less data, and becomes quite uncorrelated in this larger data setting. We believe this advocates for the design of penalties that are less reliant on distributional information concerning the separate Gaussians in the ensemble, as these penalties appear sensitive to the quality of their estimated distributions. We observe that MOPO, MOReL and the ensemble penalties perform broadly similarly despite their analytically different forms. We do observe, however, the ensemble measures display noticeable improvement as a ranking statistic. We also observe a significant loss in performance between the Transfer and Ground Truth HalfCheetah settings, with the latter being relatively poor. This implies further work is needed to develop penalties that can successfully detect the type of dynamics discrepancies that actually occur in offline MBRL. Finally, we observe that despite the similar rank correlations $\rho$, the bivariate correlations $r$ can vary considerably, and observe from the scatter plots that MOPO exhibits low kurtosis, having large penalty values 'bunched' at its extreme; we provide 3rd and 4th order moment statistics to facilitate comparison in App. C.

# 5 Key Hyperparameters in Offline MBRL

In order to design an effective search space for penalty comparison experiments, we need to understand the impact of different hyperparameters on the uncertainty estimation process itself. Furthermore, this analysis will prove useful in understanding what is important when designing these penalties in the first place.

## 5.1 How Many Models Do We Need?

Since we may have a larger compute budget due to zero experience collection in the environment, it may not make sense to copy the existing approach, originally developed for the online case where online runtime may be an issue; for instance, we can choose to train many more ensemble members. Concretely, MBPO (and subsequently MOPO) trains 7 identical probabilistic dynamics models (with different initializations). Then, when training the policy, it generates trajectories using the top 5 models based on validation accuracy, referred to as "Elites" in the Evolutionary community [31]. The reason or justification for this is not discussed in either paper, but it has seemingly been adopted in the wider MBRL setting [42, 33, 39]. In this section we seek to understand what the impact of varying this away from the default values has on the performance of the penalties discussed above.

### 5.1.1 How Does Penalty Distribution Change with Model Count?

We now vary the number of models used in the calculation of the penalties and plot their respective distributions; an illustrative example is shown in Fig. 3 with full results in App. B. The scaling of

the penalties relying on max over sets (i.e., MOPO and MOReL) is most affected as we increase the number of models due to admitting more extreme values, and we observe that the distribution shape of MOPO changes significantly as we admit more models, which we validate in App. C. This clearly impacts the ease by which we can tune this hyperparameter, as we have to contend with a changing metric distribution along with calibration quality (something we explore in the next section). Finally, we observe that simple ensemble deviation and variance change the least with differing numbers of models, highlighting their ease in tuning; this is clearly a desirable property for designing such metrics going forward.

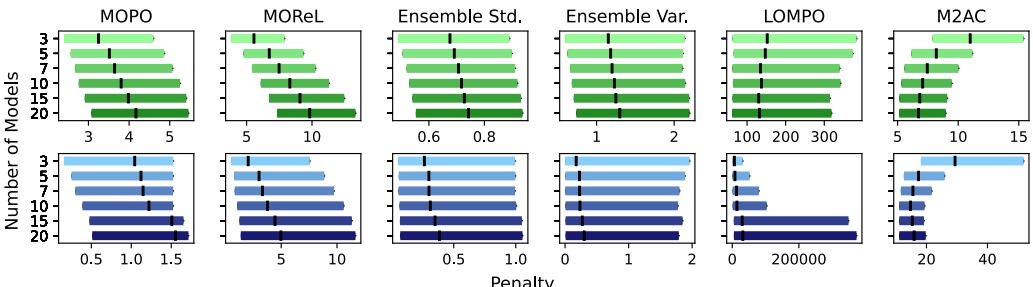

Figure 3: Box Plots showing D4RL Medium transferred to Random. We show the IQR limits and the median value denoted by the black vertical line. Green = HalfCheetah, Blue = Hopper.

### 5.1.2 How does Penalty Performance Scale with Model Count?

Empirically, there exists an optimal number of models to use in an ensemble for model-based RL [24, 30]. Up to now, heuristics have been used to select how many models we use for uncertainty estimation, despite it being possible to use a different number of models for dynamics prediction and uncertainty estimation. For instance, in MOPO transitions are generated with 5 Elite models, but all 7 models are used to calculate the penalty. In MOReL, 4 models are used for both transitions and penalty prediction. We therefore wish to understand if there is merit to using a larger number of models for uncertainty estimation compared with next state prediction.

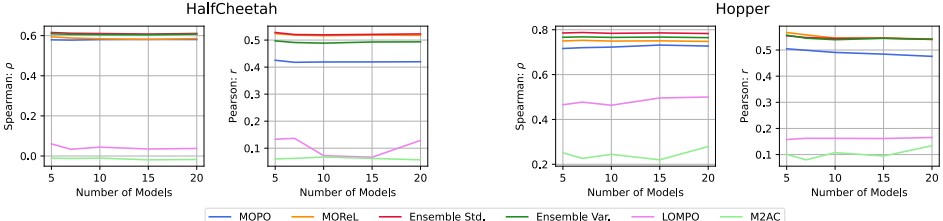

Figure 4: All Ground Truth tasks aggregated; **Left:** HalfCheetah; **Right:** Hopper

We provide a snapshot in Fig. 4, showing the aggregated results on the Ground Truth data, with full results in App. B. We see there is no clear consensus, and that the optimal number of models is highly dependent on environment, the behavior data, and penalty type, with some settings showing improving calibration with model count and vice-versa. This clearly justifies treating the number of models as a hyperparamter that is important to tune, especially on transfer tasks. Interestingly, we observe that it is possible to simultaneously improve rank ($\rho$) correlation, but reduce bivariate ($r$) correlation, especially with the MOPO penalty. This again suggests that the number of models not only affects the quality of the estimation, but also its distributional shape.

### 5.2 The Weight of Uncertainty $\lambda$

To weight penalty against reward, MOPO introduces a parameter $\lambda$ that trades off between the two terms. In their paper, the authors sweep over $\lambda \in \{1, 5\}$ for each environment. However, the optimal values may lie outside of this region, and furthermore, we have shown this value will need to drastically change to account for using a different penalty or even number of models. Clearly, this is a crucial hyperparameter for offline MBRL that needs to be tuned alongside other hyperparameters of interest.

## 5.3 The Rollout Horizon $h$

The horizon $h$ of the rollouts plays a crucial role in offline RL. Longer horizon rollouts increase the likelihood of errors in the transitions (we verify this intuition in App. D), but conversely can improve performance when errors are properly managed [19, 37]. Furthermore, as highlighted in Fig. 1, errors do not always accumulate during a single rollout in the model. Instead, we observe spikes, and note it is possible to recover from these to valid states and transitions. It is therefore imperative that a penalty identifies these spikes over the course of a model rollout and down-weights the reward accordingly.

Using this observation, we propose a novel experiment that treats these spikes as 'positive' labels, and normalize each metric to $[0, 1]$. This converts each penalty into a probabilistic classifier, and we evaluate how well they classify OOD events that occur increasingly under longer $h$. This is precisely the intuition behind the MOReL and M2AC approaches, whereby the penalty acts as an 'anomaly' detector, removing detrimental transitions that lie above a threshold. The analysis in this section can also be viewed as assessing the efficacy of penalties under these schemes, where binary detection is more important than correlation. Finally, we assess two ground truth errors: the dynamics discrepancy (as before), and also introduce the distance from the offline distribution trained on, which we measure as the 2-norm between a state-action tuple and its nearest point in the offline data; these are called 'Dynamics' and 'Distribution' respectively. We provide precision-recall curves and more details on this experiment in App. D and E.

Table 2: Performance of different penalties as OOD event detectors averaged over all datasets in Hopper and HalfCheetah. AUC is 'Area Under Curve' and AP is 'Average Precision' (higher is better for both).

| | Percentile | | | | | | | | | | | |
| | 90th | | | | 95th | | | | 99th | | | |
| | Dynamics | | Distribution | | Dynamics | | Distribution | | Dynamics | | Distribution | |
| Penalty | AUC | AP | AUC | AP | AUC | AP | AUC | AP | AUC | AP | AUC | AP |
| --- | --- | --- | --- | --- | --- | --- | --- | --- | --- | --- | --- | --- |
| MOPO | 0.886 | 0.503 | 0.759 | 0.345 | 0.893 | 0.351 | 0.800 | 0.273 | 0.921 | 0.200 | 0.885 | 0.157 |
| MOReL | 0.897 | 0.537 | 0.774 | 0.343 | 0.905 | 0.403 | 0.814 | 0.279 | 0.931 | 0.260 | 0.886 | 0.148 |
| Ensemble Std. | 0.902 | 0.551 | 0.794 | 0.378 | 0.907 | 0.401 | 0.834 | 0.309 | 0.929 | 0.251 | 0.904 | 0.177 |
| Ensemble Var. | 0.903 | 0.559 | 0.777 | 0.352 | 0.910 | 0.419 | 0.817 | 0.287 | 0.933 | 0.270 | 0.891 | 0.158 |
| LOMPO | 0.662 | 0.328 | 0.735 | 0.326 | 0.673 | 0.211 | 0.760 | 0.250 | 0.731 | 0.088 | 0.805 | 0.111 |
| M2AC | 0.585 | 0.206 | 0.676 | 0.235 | 0.597 | 0.115 | 0.696 | 0.140 | 0.650 | 0.039 | 0.717 | 0.048 |

We observe that the penalties are powerful at identifying dynamics discrepancy, but not as accurate at identifying when the world-model data is out-of-domain with respect to the offline data. This is a well known phenomenon in deep neural networks and has been recently investigated in terms of feature collapse [45], where latent representations of points far away in the input space get mapped close together. On the other hand, this shows an important distinction between the regularization induced by MBRL uncertainty and explicit state-action regularization in model-free approaches, such as [47, 23]. In the latter approaches, policies are penalized for taking out of distribution actions w.r.t. the offline dataset, but this is not always the case with policies trained under MBRL and uncertainty penalties. The success of MBRL methods in RL may therefore lie in the generation of state-action samples that are OOD but represent accurate dynamics, thus facilitating dynamics generalization in policies; recent work has shown that specifically augmenting dynamics without taking into account state-action shift can improve offline RL policy generalization OOD [4]. We believe future work understanding the implications of this property is vitally important.

## 5.4 Implementation Details

The above discussion captures many of the key *hyperparameters* specific to current offline MBRL algorithms. However, there are significant *code-level* implementation details which are often critical for strong performance and make it hard to disambiguate between algorithmic and implementation improvements. Worryingly, many of these details are not mentioned in their respective papers, or are different between the authors' code and paper. We detail clear examples of this in App. F. We believe further investigation of these code-level implementation details represents important future work, as has already been done for policy gradients [12, 1]. Indeed – it is unclear if the improvement of MOReL over MOPO is due to its P-MDP formulation or if it is successful *in spite of* this formulation, due to a superior policy optimizer or dynamics model design. We believe that this paper takes a

significant first step in tackling this issue by directly comparing a number of proposed penalties along with other important implementation factors and understanding their individual impact.

# 6 Testing the Limits of Current Approaches

In this section we seek to answer the following question: how well can existing methods perform, given optimal selection of the discussed hyperparameters? To answer this question, we use a state-of-the-art Gaussian Process-Bayesian Optimization (GP-BO) algorithm, CASMOPOLITAN [46], and tune the configuration for each individual environment. Each BO iteration is run for 300 epochs on a single seed. CASMOPOLITAN uses tailored kernels and trust regions to handle mixed categorical and continuous hyperparameter search spaces. The hyperparameters are listed in App. G. We define our search space over:

- **Penalty type (categorical):** taking values over {MOPO, MOReL, LOMPO, M2AC, Ensemble Std, Ensemble Variance}.
- **Penalty scale $\lambda$ (continuous):** taking values over $[1, 100]$.
- **h (integer):** taking values over $\{1, 2, \ldots, 50\}$.
- **Models N (integer):** taking values over $\{1, 2, \ldots, 15\}$.

Our implementation mimics MOPO in that we use the same probabilistic dynamics models (with unchanged hyperparameters) and policy optimizer (SAC, [17]), which differs from MOReL which uses Natural Policy Gradient [20]. The focus of our experiment is to explore parameters relating to *uncertainty quantification*, and we believe this is a sufficiently fair set up.

Table 3 shows the optimal discovered hyperparameters. We note that the only penalties chosen are the MOPO and ensemble penalties, corroborating the findings in our analysis that these are often the most effective. We observe that MOReL is not chosen, likely because ensemble penalties are generally better correlated with true dynamics error, and are easier to tune since their scaling changes less with increasing model number; we also observe that MOReL has very similar shape statistics to Ensemble Std. (App. C).

Table 3: Optimal discovered hyperparameters using BO

| Environment | | N | $\lambda$ | h | Penalty |
|---|---|---|---|---|---|
| | | | | | **Discovered Hyperparameters** |
| HalfCheetah | random | 10 | 6.64 | 12 | Ensemble Std |
| | mixed | 11 | 0.96 | 37 | Ensemble Variance |
| | medium | 12 | 5.92 | 6 | Ensemble Variance |
| | medium-expert | 7 | 4.56 | 5 | MOPO |
| Hopper | random | 6 | 4.46 | 47 | Ensemble Std |
| | mixed | 7 | 5.90 | 5 | MOPO |
| | medium | 7 | 20.03 | 31 | Ensemble Std |
| | medium-expert | 12 | 39.08 | 43 | MOPO |

The selection of MOPO is also explainable; we observe it displays significantly lower skew and kurtosis than all other metrics (App. C), whilst still maintaining competitive rank correlation. We also found that in all Hopper experiments, Ensemble Var. never achieved high performance, and its only major difference to Ensemble Std. lies in its distributional shape. Interestingly, in HalfCheetah, the opposite is true, with Ensemble Var. delivering significant performance gains. This implies that distributional shape may play as important a role as overall calibration, and advocates for the learning of *meta-parameters* that control for these.

We note that values of the rollout horizon $h$ and penalty weight $\lambda$ differ greatly from those chosen in the original MOPO paper, which chooses from $\{1, 5\}$. Notably, the Hopper environments prefer a much longer rollout length and higher penalty weight, even accounting for the magnitude of the penalty used. Again this is backed up by our analysis; along a single rollout dynamics errors do not necessarily accumulate, they simply become more likely to occur. As long as we penalize the aforementioned spikes appropriately, we can handle longer rollouts, and generate more on-policy data. The number of models used to compute the uncertainty estimates can also differ greatly from the standard 7. This again aligns with our findings that using more models for uncertainty estimation can be beneficial, but is dependent on environment, data, and penalty.

Table 4: Comparative evaluation on the D4RL benchmark suite against other model-based RL algorithms. The raw score for Optimized (Ours) and MOPO (Ours) was taken to be the average over the last 10 iterations of policy learning averaged over 4 seeds. Results of MOPO and COMBO were taken from the COMBO paper. Results for MOReL were taken from its paper.

| Environment | | Optimized (Ours) | MOPO (ours) | MOPO (authors) | MOReL | COMBO |
|---|---|---|---|---|---|---|
| HalfCheetah | random | 31.7 | 32.7 | 35.4 | 25.6 | 38.8 |
| | mixed | **58.0** | 52.8 | 53.1 | 40.2 | 55.1 |
| | medium | 45.7 | 46.5 | 42.3 | 42.1 | 54.2 |
| | medium-expert | **104.2** | 67.6 | 63.3 | 53.3 | 90.0 |
| Hopper | random | 12.1 | 4.2 | 11.7 | 53.6 | 17.8 |
| | mixed | 90.8 | 66.7 | 67.5 | 93.6 | 73.1 |
| | medium | 46.5 | 17.3 | 28.0 | 95.4 | 94.9 |
| | medium-expert | 105.8 | 24.9 | 23.7 | 108.7 | 111.1 |

Table 4 how these unconventional hyperparameter choices fare against state-of-the-art offline model-based RL algorithms. We include a comparison of our implementation of MOPO v.s. the authors' reported performance using the same hyperparameters. We note the two are relatively similar and thus we are able to make a faithful comparison. Our method, which we label as "Optimized (Ours)", is state-of-the-art on the Halfcheetah mixed and Halfcheetah medium-expert environments by a strong margin. Further notable results include the hopper mixed and hopper medium-expert environments which show we are able to tune a MOPO-like method up to the performance of COMBO and MOReL. The importance of good uncertainty quantification and hyperparameter selection for MOPO is illustrated in Fig. 5 where we show we can improve MOPO performance by over 5x whilst obtaining a stable solution.

Limitations of our work include the fact that we solely performed BO over the hyperparameters which directly had an influence on *uncertainty quantification*. Other hyperparameters which have a significant general impact on MBRL performance include the number of Elites and the model training hyperparameters [50] (i.e., learning rate, weight decay). Each BO iteration evaluated a hyperparameter setting on a single seed which could introduce stochasticity; we do however expect the Gaussian Process surrogate model to account for this aleatoric uncertainty. We also note that individually fine-tuning hyperparameters for each environment is not tractable; due to this we only performed BO over 2 environment types in the D4RL suite. However, the same method could be used to find an optimal single configuration for *all* environments. We also use true environment reward as BO feedback, whereas in reality we may be forced to use offline/off policy evaluation (OPE) [29, 15]. However we do note that our solutions can be more stable over policy training iterations than previous works, and we believe that metrics useful for training will also be useful for direct method OPE.

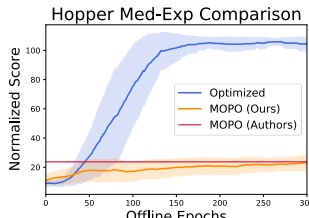

Figure 5: Comparison of MOPO performance on the Hopper medium-expert environment.

The primary goal of our work is to improve understanding of existing methods, the majority of which we believe will be used for good. Indeed, offline RL promises to be beneficial in a variety of real-world settings, such as healthcare [40] and robotics [11]. However, we note that it is of course possible our findings aid those looking into applying these methods for malicious use.

## 7  Conclusion

In this paper, we rigorously evaluated the impact of various key design choices on offline MBRL, comparing for the first time a number of different uncertainty penalties used in the literature. By proposing novel evaluation protocols, we have also gained key insights into the nature of uncertainty in offline MBRL that we believe will be of benefit to the wider RL community. We demonstrated the impact of this analysis by improving upon existing offline MBRL methods in performance with significant changes to key hyperparameters compared to prior work, obtaining significantly improved performance in almost all benchmarks.

Going forward, we are particularly excited by developments in offline/off-policy evaluation [15, 7] to facilitate accurate assessment of agent performance without querying the environment. This would then open the door for population-based training methods [18, 38], which have shown great success in online MBRL [50]. Furthermore, throughout the paper we have highlighted potential areas of interest, from better understanding the role of implementation details, through to the development of meta-parameters controlling penalty distribution shape attributes.

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
