# OpenReview forum: "Revisiting Design Choices in Offline Model Based Reinforcement Learning"
_NeurIPS.cc/2021/Conference — NeurIPS 2021 Submitted_

### Official Review · Reviewer_Zw3Q · 2021-07-14

**Rating:** 6
**Confidence:** 3

**Summary:**

This paper considers the problem of offline, model-based reinforcement learning, wherein a dynamics model is learned from offline data, and subsequently, simulated trajectories in this learned model are used to train a policy. A major challenge in model-based offline RL is how to handle the inevitable distributional shift that occurs when the learned policy is deployed on the true system. Previous works have proposed to mitigate this issue by considering a pessimistic MDP, wherein the reward function is augmented by a measure of the model mismatch between the estimated and true dynamics to obtain an MDP on a which a policies expected return is a lower bound for its expected return on the true dynamics.

This paper compares a collection of prior work following this general approach, but differing in hyperparameters such as the formulation of the model mismatch penalty, rollout horizon, and number of models used the ensemble dynamics model. The paper applies Bayesian optimization to offer insights as to which choices are most effective on two test domains, the Hopper and Half-Cheetah.

**Limitations And Societal Impact:**

The discussion of societal impacts and limitations is reasonable given the content of this work. The limitations are clearly discussed, and, as this work is focused on experimental comparisons of existing techniques in model-based offline RL, the short discussion of potential negative societal impact was adequate.

**Main Review:**

This paper addresses an important topic in the model-based RL literature by comparing several related techniques on a common implementation, and performing a suite of experiments to highlight the importance of various hyperparameters common to these algorithms. Work to this end is significant, as it provides empirical insights to inform future algorithm development. Overall, I think this work presents a good set experimental results, but I feel that there are some limitations in the clarity of the message / presentation of the results as well as limitations in the scope of the experiments that hold this paper back.

First, while the authors do a good job of covering related work and background, I found that some sections were hard to follow, including the description and discussion of the experiments. This left me with several questions:
	- 4.2: How exactly was the degree of exploitation of the policy controlled? The specific generation of this policy should be explained in the main paper.
	- 4.2: The name "ground-truth" for this experiment is confusing, as I'm not sure how this experiment is any more "ground-truth" than the "transfer" experiment.
	- 5.1.2: The experiment measures the quality of a particular choice in terms of correlation coefficients, but this choice of optimality measure is not well motivated. The authors should either plot MBRL performance with all hyperparameters fixed apart from the number of models, or provide more theoretical justification why higher correlation should lead to better performance.
	- 5.3: The section is titled Rollout Horizon, but the bulk of it is spent discussing an interpretation of the dynamics penalty as an OoD detection scheme, investigating the utility of these penalties as binary classifiers. The details of this experiment are unclear, however: what data was used to train the model, and what data was used to evaluate the penalties? (other sampled transitions or the transitions from rollouts of a learned policy?) What does "percentile" indicate in Table 2? From the appendix, I see that this corresponds to the error threshold used to deem points as OoD, but this should be explicitly detailed in the body of the paper.
	- What are the differences the different experimental domains ("random", "mixed", etc.)? The authors should describe the experimental domains clearly in the body of the paper.

Overall, the authors should present the experiments with greater clarity, making it clear how each experiment was carried out, motivating the choice of metric for each experiment, and clearly defining terms and notation.

Second, the experiments were limited, making it hard to understand how well these results would generalize. The authors perform these studies on only two domains, both MuJoCo based environments with dense reward functions. The penalty function as motivated by the PMDP formulation has a key connection to the value function of the MDP, and not just the dynamics, so it could be worthwhile to perform these studies with different reward functions. Furthermore, the Bayesian optimization experiments only show results of performance on the same environment that was used to optimize the hyperparameters. While this answers the question that the authors pose -- "how well can we do if we choose optimal hyperparameters?" -- this does not provide any actionable insights into the more practical question of "what hyperparameters should I use on a new environment?" Results on how well the optimized parameters transfer between environments would provide more insight and strengthen the paper.

Overall, this paper represents a good step towards the important goal of strong empirical comparisons of recent offline MBRL techniques, but it is hindered by the limitations in exposition and the discussion of the experiments.


Minor comments:
	- Notation in equation 1 is not defined (what is $eta_M$?). If $eta_M$ corresponds to the expected discounted return, as it does in [49], then this equation gives an upper bound, rather than a lower bound which is written in the text. However, it appears that in [49] the inequality is flipped, so perhaps this is a typo.
	- The best performing methods should be bolded in all tables. Confidence intervals should be included for tabular results as well.
	- The inline table in sections 4.2  is confusing, as it contains results from the experiment described in section 4.1. This should be a floating element and not inline.

**Time Spent Reviewing:**

3

---

> ### Author Response · Authors · 2021-08-09
> **Response to Reviewer Zw3Q**
>
> We appreciate the reviewer acknowledging the transferability of our insights to future work, and thanks to their comments, we believe we have significantly improved our work. If there is more we need to clarify, please let us know, otherwise we kindly request the reviewer considers raising their score.
> We will now address each concern individually:
>
> 1. *The specific generation of this policy should be explained in the main paper*
>
> We believe this was addressed in the main paper in lines 173-178, but agree that lower level details are missing, such as number of exploitation policies chosen, as well as the number of offline epochs we chose to train them for. Concretely, we selected the 5 most ‘exploitative’ models (i.e., the models which gave the largest difference in model return $-$ actual return) over a set of 1000 checkpoints (i.e., we trained the policies offline for 1000 epochs each). We will include these details in the appendix.
>
> 2. *The name "ground-truth" for this experiment is confusing*
>
> This criticism is fair, and we concede that the terminology is opaque in retrospect. Concretely, ground-truth refers to the fact that the dynamics MSE we calculate is with respect to the ‘ground-truth’ environment, despite featuring state/action tuples that are generated inside the model. With this in mind, we believe changing the terminology to “True Model Based” should help resolve this confusion.
>
> 3. *This choice of correlation as an optimality measure is not well motivated*
>
> We disagree with this point, as we believe the experimental part of the paper precisely motivates these metrics, especially taking into account the new results. For instance, we show that the LOMPO and M2AC penalties are consistently very poorly correlated (both rank and bivariate wise) with dynamics errors, and hence the BO algorithm *never* selects these. Conversely, the more strongly correlated Ensemble-based penalties are selected most frequently, justifying the use of correlation as a proxy for downstream performance.
>
> 4. *Some details should be in the main body*
>
> We accept that some details require reference to the appendix, such as the definition of percentile, how the data was generated for the ROC/AUC experiments (this was done using the same exploitative policies + ground-truth approach as before), and will update the paper’s main body to elucidate these points. However, some details, such as which data we trained for the ROC/AUC analysis, are in the main body (“...averaged over all datasets”). We believe that the D4RL dataset is well established in the offline literature, but concede that to those unfamiliar with offline RL, the nomenclature may be obscure. We will therefore add a line to make it clear where the terms such as “random” and “expert” are derived.
>
> 5. *The experiments were limited… doesn’t answer a more practical question*
>
> We fully accept this criticism, and as a result of the reviewer’s feedback, we have run significantly more experiments on other domains to justify the existing analysis and concretely provide answers to the practical question. As we show in the table within our general feedback, our experiments on a variety of other offline domains demonstrate that our findings generalize to all new settings, including ones that are significantly different in nature and, to our knowledge, *never previously benchmarked in offline MBRL*. Through the extensive new experiments, we believe we adequately answer the practical question, namely that we ought to favor: 1) algorithm designs and hyperparameter choices that utilize longer horizons; 2) well calibrated uncertainties; 3) relatively larger penalty coefficients. We would also like to add that this is fully supported by the fine-grained analysis that precedes it (e.g., Sections 4 and 5), making the link much clearer between these two sections, and showing the empirical validation of these preceding insights.
>
> 6. *Minor comments*
>
> The reviewer is indeed correct regarding the flipped sign, we will change this in the updated manuscript! We will also endeavour to incorporate the suggested formatting changes and include confidence intervals for our tabular results.
> Thank you!

---

> > ### Comment · Reviewer_Zw3Q · 2021-08-31
> > **Thanks for the response!**
> >
> > 1. I agree that lines 173-178 address this, but as policy selection is a key part of your experimental evaluation, it would be good to make this more explicit in how the original set of policies was produced - did you train one policy and select among different model checkpoints? or train multiple policies to convergence from different initializations or random seeds? Training from different seeds seems more intuitive to fully explore the space of exploitative policies..
> >
> > 2. Thanks, the proposed name is better.
> >
> > 3. Justifying the choice of metric via experimental evidence of the link between correlation coefficient and final MBRL performance would be fine, but (1) this should be discussed in the text (as this section comes before the Bayesian optimization results), and (2) plots showing MBRL performance against the observed correlation of the penalty would be more convincing than the observation that a black box Bayesian optimization model tends not to choose penalties with low correlation.
> >
> > 4. Thanks, I think this will make the paper more readable for a wider audience.
> >
> > 5. I appreciate the additional experiments. The fact that these experiments support a concrete recommendation addresses my concern.

---

> > > ### Author Response · Authors · 2021-09-02
> > > **Response to further questions**
> > >
> > > Thanks so much for your response and further questions:
> > >
> > > 1. We trained our policies using 3 random starting seeds, and selected the most 5 exploitative from this set of policies; we found in general that we would get fairly good coverage (e.g., most of the times we'd be selecting policies from all 3 seeds). We will include these details in the appendix.
> > >
> > > 3. Thanks for raising this point - we have now optimised for each individual penalty for the Hopper environment so we can see MBRL performance against the observed correlation of the penalty.
> > >
> > > | Environment | Dataset type | Ensemble Var | Ensemble Std | Max Aleatoric | LL Var | LOO KL Div | Max Pairwise Diff |
> > > |:---:|:---:|:---:|:---:|:---:|:---:|:---:|:---:|
> > > | hopper | random | $\textbf{376.08}$ | 373.29 | 363.56 | 61.07 | 218.16 | 346.43 |
> > > | hopper | mixed | 2340.68 | $\textbf{2818.28}$ | 1787.42 | 40.09 | 57.42 | 1855.95 |
> > > | hopper | medium | 592.92 | $\textbf{2235.12}$ | 1955.51 | 10.32 | 170.63 | 1321.67 |
> > > | hopper | med-exp | 1065.21 | 2275.41 | $\textbf{3483.87}$ | 7.67 | 2009.86 | 2924.45 |
> > >
> > > We will include this discussion in the main paper.
> > >
> > > Thanks so much again for your comments which will help improve our paper, please let us know if we can clarify further!

---

### Official Review · Reviewer_mit8 · 2021-07-19

**Rating:** 5
**Confidence:** 5

**Summary:**

Offline RL has seen an explosion of interest in the past year or so, with model-based RL methods being the state of the art (or close to it) on commonly studied benchmark tasks (like D4RL and RL Unplugged). This paper takes a retrospective view on model-based offline RL to benchmark/evaluate different design choices in recent papers. Based on experimental evaluations, the paper makes a number of interesting observations and suggestions for improvements such as longer rollout horizons and penalties more closer to OOD detection rather than uncertainty estimation, among others.

**Limitations And Societal Impact:**

There are quite a few limitations, some of which were acknowledged in the paper. I have provided more detailed comments in the main review.

**Main Review:**

**Strengths**
- Important domain for retrospective study considering broad recent interest.
- Very interesting choice of research questions and design methodology for studying them.

**Weaknesses/Limitations**
- While the questions studied are interesting, I find that many major design factors were not studied. In particular, the choice of the policy optimization between MOReL (NPG/TRPO) and MOPO (SAC) are different. In my mind, this is a major point of departure between the methods and can explain much of the differences -- NPG/TRPO in general works well on Hopper environments, while SAC substantially outperforms it in HalfCheetah. We see the same trend reflected in MOReL vs MOPO too.
- MOReL also has a penalty term when truncating rollouts due to OOD detection. I don't see this as part of the implementation or study in Sections 5 and 6. Can the authors clarify?
- I disagree with lines 306-309 in Section 6. In my opinion the policy optimization algorithm and uncertainty estimation are inseparably intertwined. More concretely, in MOReL, the policy is initialized with behavior cloning and updated by taking small NPG/TRPO steps that ensures the policy does not deviate too much. Thus, the space over which the policy searches may not stray too far from the data distribution by design enabling certain types of penalty schemes to work better than others. MOReL also implements much of the final suggestions in the paper -- it uses a much longer rollout horizon of 500-1000, uses discrepancy more for OOD detection than uncertainty estimation etc. By making fundamental design changes to how MOReL is actually implemented, the retrospective study is unfair in its comparisons.
- Section 6 appears to be a bit misleading. Ensemble std/var was not used in any prior work, somewhat creating a departure from a pure retrospective study (it mixes up comparing older algorithms with providing a newer one). This leads to a skewed results in favor of other methods due to high correlation between MOReL and ensemble std/var. I would suggest the authors to alter the presentation -- present the results/scores for the best hyperparameters for each penalty category. This will likely provide a more comprehensive and unbiased retrospective.
- [Minor] The appendix mostly just contains plots and tables without providing much context on the experiments. Adding more text and clearly explaining each figure and table can be helpful.

**Final summary** Overall, the paper studies a very important and timely topic given the explosion of interest in offline RL and proliferation of model-based methods. That said, for me, retrospective papers need to achieve a high bar in terms of being thorough, unbiased, and having a very clean presentation, because it is very hard to write a retrospective paper on another retrospective paper. While some of the limitations are acknowledged in the paper, they are nevertheless limitations and must be addressed (and can be addressed with more time and experiments).

Ultimately, in my opinion the shortcomings of the study are more than the strengths, and thus I am unable to recommend acceptance at this stage. I would recommend the authors to incorporate suggestions from my review (and others), expand the set of design choices studied, and improve the submission for a future conference cycle.

**Time Spent Reviewing:**

3

---

> ### Author Response · Authors · 2021-08-09
> **Response to Reviewer mit8**
>
> We appreciate the comments of the reviewer, which we have taken on board in changes to our work. We believe the reviewer missed the primary purpose of our study, for which we are to blame. We want to emphasize that this study is not about the MOReL *algorithm*, but solely about the *choice of uncertainty penalty*. In particular, we think the behavior cloning/NPG choice is orthogonal, as a better uncertainty penalty will still improve with these other elements. Concretely, there is nothing inherent to the MOReL max pairwise discrepancy penalty that makes it particularly suited to an approach that features BC/on-policy/NPG; it is simply a proxy for the dynamics total variation distance. We therefore wish to analyze both the behavior and statistical properties of the *penalty* that was introduced in MOReL, not the method itself. Indeed our analysis shows that it can be *statistically indistinguishable* to our Ensemble penalties, whilst having lower correlation with the true error and being more sensitive to design choices such as ensemble size.
> Moving on to more specific comments:
>
> 1. *In my opinion the policy optimization algorithm and uncertainty estimation are inseparably intertwined*
>
> When designing an algorithm, it is clear that the impact of being OOD can be further ameliorated through steps such as behavior cloning and conservative policy updates. However, this paper is not about designing a SoTA algorithm. It is about seeing which penalty most accurately captures uncertainty with offline data and its impacts on downstream performance. This should be agnostic to the policy optimizer, especially in light of our findings in Section 4 and the Appendix regarding observing the similarity of these metrics through the lens of their descriptive statistics. Furthermore, while we agree that the policy optimizer can impact performance in certain environments, it is difficult to disentangle why MOReL performs well on Hopper, as they evaluate on **D4RL v2 environments**, [where it is known that the Hopper data has been changed](https://github.com/rail-berkeley/d4rl/issues/86), with some offline trajectories (especially `medium`) being higher in average return. However we stress this is an orthogonal point; we aren’t claiming one P-MDP formulation is better than the other, we are simply understanding how design choices *common to both* can affect offline MBRL performance.
>
> 2. *MOReL uses 500-1000 step rollouts.*
>
> This is true, however it is also an upper bound because MOReL has an early termination criteria; to our knowledge the authors never show how long the rollouts performed in the model *actually* were. Indeed, our work indicates that this termination may not be required, since the spikes in model error are only temporary. Even casting this aside, we believe our insights are complementary to the horizon hyperparameters in MOReL. Current consensus is unclear whether longer rollouts in offline MBRL are beneficial/appropriate, with MOPO electing a very conservative 5 step horizon, in comparison to the 500-1000 of MOReL. We therefore believe our analysis provides clear and novel insight into the repercussions of longer rollouts, both on performance, as well as the manifestation of errors in the model. We believe this is important for future algorithm design in this area, and MBRL more generally.
>
> 3.  *The study is unfair because we changed components to MOReL.*
>
> We again emphasize **this is not a comparison about MOReL vs. other algorithms**. It is about comparing the uncertainty penalty used and their impact on other design choices. It is clear that overloading the names of the algorithms to *also mean their penalties* could cause significant confusion. Therefore in light of the reviewer’s feedback, we have renamed the penalties as follows: MOPO -> Max Aleatoric; MOReL -> Max Pairwise Diff; LOMPO -> LL Var (Log-Likelihood Variance); M2AC -> LOO KL (Leave-One-Out KL Divergence). We have now reflected this in the new results, and **will change this throughout the paper**. We hope this makes the aims of the paper clearer.
>
> 4. *Section 6 appears to be a bit misleading… not retrospective...*
>
> We restate that this paper does not aim to be a retrospective in the sense of performing a comparison between existing algorithms, and the offline MBRL field may be too immature for this to be done satisfactorily at this point. Instead, we focus on providing insights that should: 1) aid in future offline MBRL work; 2) help us better understand the problem setting by conducting novel analysis into the precise impact of design choices common to most methods; 3) highlighting under-used elements from Baysian deep learning that are beneficial for RL researchers going forward. Indeed, we strongly believe the insights we gain here could be used to *help improve MOReL*, such as showing the careful balance between model number and penalty scaling (and shape statistics), the choice of penalty itself, as well as the analysis of early termination v.s. continuing to rollout past this point. Indeed, as stated in the paper, we strongly believe the AUC/ROC analysis performed is *directly relevant* to the design of MOReL-style algorithms, whereby the uncertainty is used as a threshold for a dynamics anomaly detector, thus acting as a binary classifier for such events.
>
> 5.  *Present the results/scores for the best hyperparameters for each penalty category*
>
> The following table presents the best found single-seed performance by the BO algorithm split by each penalty category on the hopper environment. We hope this provides a more comprehensive picture on where each penalty stands.
>
> | Environment | Dataset type | Ensemble Var | Ensemble Std | Max Aleatoric | LL Var | LOO KL Div | Max Pairwise Diff |
> |:---:|:---:|:---:|:---:|:---:|:---:|:---:|:---:|
> | hopper | random | $\textbf{376.08}$ | 373.29 | 363.56 | 61.07 | 218.16 | 346.43 |
> | hopper | mixed | 2340.68 | $\textbf{2818.28}$ | 1787.42 | 40.09 | 57.42 | 1855.95 |
> | hopper | medium | 592.92 | $\textbf{2235.12}$ | 1955.51 | 10.32 | 170.63 | 1321.67 |
> | hopper | med-exp | 1065.21 | 2275.41 | $\textbf{3483.87}$ | 7.67 | 2009.86 | 2924.45 |
>
> Overall we thank the reviewer for their comments that have already significantly improved our paper. In light of these changes, and the new experiments (see the general comment) we hope the reviewer now feels the paper is ready for *this* conference cycle and can raise their score. Thank you!

---

### Official Review · Reviewer_hd7x · 2021-07-19

**Rating:** 6
**Confidence:** 4

**Summary:**

The authors introduce several assesment strategy to evaluate risk-sensitive criterions in model-based RL. In particular the authors focus on more recent proposal to incorporate risk into model-based RL and how these criteria interact with other hyper-parameter of model-based RL such as the roll-out horizon.

**Limitations And Societal Impact:**

yes.

**Main Review:**


There are several things I like about this:

1. The paper addresses a relevant issue: How to incorporate risk in model-based (particularly, offline (or batch) model-based RL is still an open question

2. The experiment make sense given the research question.  This is the first paper I am aware that tries to systematically evaluate these different risk criterions

3. The experiments are through, while only two benchmarks are considered (see my cons) the authors investigate the different risk-sensitive criterions on different, and interesting aspects.

Cons:
1. Some evaluations, such as 4.2  do not appear to be a novel approach. To my knowledge evaluating predicted returns in the model with the ground truth is a standard process in model-based RL.  This is hard to asses, as such evaluation are often done in practice but may have not been formulated in clarity and as a reproducible performance indicator in literature so far.

2. The choice of environments in particular is limited to HalfCheetah and Hopper which make it difficult to asses the universality of the finding. Additionally, these benchmarks to my knowledge do not exhibit any intrinsic stochastic dynamics.  Risk-sensitive criterions that include aleatoric uncertainty may not be suited for these type of problems. If they are, these improvements may be due to heuristic reasons (aleatoric models may be more smooth regressors which generalize better?)

3. Relevant literature is missing. E.g on risk-sensitive criterions [1], on batch (offline) RL [2]  on uncertainty-based risk-sensitive RL [3,4,5,6].

- [1] Garcıa, Javier, and Fernando Fernández. "A comprehensive survey on safe reinforcement learning." Journal of Machine Learning Research 16.1 (2015): 1437-1480.
- [2] Lange, Sascha, Thomas Gabel, and Martin Riedmiller. "Batch reinforcement learning." Reinforcement learning. Springer, Berlin, Heidelberg, 2012. 45-73.
- [3] Berkenkamp, Felix, et al. "Safe model-based reinforcement learning with stability guarantees." arXiv preprint arXiv:1705.08551 (2017).
- [4] Depeweg, Stefan, et al. "Decomposition of uncertainty in Bayesian deep learning for efficient and risk-sensitive learning." International Conference on Machine Learning. PMLR, 2018.
- [5] Sun, Yi, Faustino Gomez, and Jürgen Schmidhuber. "Planning to be surprised: Optimal bayesian exploration in dynamic environments." International Conference on Artificial General Intelligence. Springer, Berlin, Heidelberg, 2011.
- [6] Bagnell, J. Andrew, Andrew Y. Ng, and Jeff G. Schneider. "Solving uncertain Markov decision processes." (2001).


Overall, I believe this paper serves as nice addition to the literature, as it develops reproducible performance indicators for offline model-based RL, and connects these to the relevant hyper-parameters of this approach, such as the roll-out horizon. Therefore I am leaning towards acceptance, however,  the limited (and deterministic) choice of benchmarks makes me have doubts about the universality of the findings.


**Time Spent Reviewing:**

3

---

> ### Author Response · Authors · 2021-08-09
> **Response to Reviewer hd7x**
>
> Thank you for your time and for highlighting the relevance of our analysis to ongoing research into offline model based RL. Please see the new results in our general response and specific comments below. We believe we have significantly improved our work and kindly request the reviewer consider raising their score. If there is more we need to clarify, please let us know.
>
> 1. Experimental evaluation
>
> While it is true that difference between real and model-based *returns* is not a novel metric (indeed this is a common benchmark in OPE papers), our approach of generating the ground truth transition difference is in fact novel. As highlighted in the paper, along a single rollout inside the model, we also replay the *exact* same state and action inside the MuJoCo simulator, and compare that *true next state* to the prediction from our model. This is novel for three reasons:
> - It assesses the true dynamics modelling error compared with the *ground truth* simulator at *each time step*, not just the aggregate return, allowing us to determine whether the uncertainty penalties can truly capture these errors at *time step level granularity*.
> - We are assessing the uncertainty penalties under *covariate shift*, because our state and actions now come from rolling out our policy *inside the model* (i.e., planning), which produces states and actions that did not occur in the D4RL offline data set (i.e., the model training data). This is in contrast to previous work, such as in COMBO, which included a comparison of the correlation of penalties against MSE error, but this is done on a held-out test set coming from the *same offline data distribution* that the model was trained on and thus measures in-distribution calibration which is not the offline MBRL problem setting.
> - This shows, for the first time, the precise nature of how errors accumulate inside a single rollout (e.g., sudden spikes), and also how models can in fact *generalize dynamics* outside the model training data, something also relevant to the wider MBRL community.
> Regarding performance indicators, we show experimentally that penalties which score better under this evaluation are also selected more often by the BO method when optimizing for stability and expected return, and by corollary the worst performing metrics are never selected and discarded early.
>
> 2. Experimental Limitations (2 Benchmarks, stochastic dynamics)
>
> We agree this *was* a limitation of our work. In light of the reviewer’s feedback, we have significantly increased the scope of our final experiments. Firstly, we have added a *third D4RL locomotion domain* (Walker2d). Secondly, we also added a completely different set of tasks from the *manipulation domain*, which to our knowledge are a first for offline MBRL and features *sparse rewards*. Please see above for more details. Concretely, we believe these serve as very strong empirical validation of the novel insights we provide in Sections 4 and 5 of the paper.
> To our knowledge, none of the methods in the offline MBRL literature have considered stochastic dynamics (indeed none of the environments in D4RL are stochastic), although the aleatoric heads of the model are designed to deal with precisely this type of data. We will mention this in the paper, and it certainly represents an interesting future research direction. We believe that this is out of the scope of this paper because the impact of risk-sensitivity/uncertainty penalization (especially for rollouts performed inside the model) for even standard deterministic environments (predominant in continuous control) has been poorly understood up to now. In short, we believe it is important to understand the intricacies and pathologies of *existing design choices on common environments* before delving more deeply into less common benchmarks.
>
> 3. Missing literature
>
> We thank the reviewer for highlighting these very interesting works. We will add these to the paper and include an appropriate discussion.
>
> Thank you again!

---

### Author Response · Authors · 2021-08-09
**General Response**

We’d like to thank the reviewers for their time. We were encouraged that the reviewers noted the importance of our work systematically analyzing important design choices in offline MBRL. A common theme amongst the reviewers was that our paper would be considerably improved with a larger experimental evaluation. We fully accept this, and are pleased to present *additional results on the D4RL Walker2d environments*, as commonly used in offline MBRL works. In addition, we also include the Adroit Pen and Hammer environments, *which as far as we are aware have not previously been used in offline MBRL*, presenting very different challenges to the MuJoCo locomotion tasks featuring *sparse rewards*, real human demonstrations and narrow data distributions.

Moving to the results, we continue to show that *carefully chosen hyperparameters are enough to outperform significantly more complex algorithms* on average across all tasks. Furthermore, we for the *first time* show strong performance for offline MBRL methods on the Adroit tasks. This very clearly demonstrates the generality of our findings in Sections 4 and 5, and paves the way for future work.

We now have a total of 18 datasets, over twice the number that the paper featured originally, and across a total of 5 different domains. Given the borderline initial scores, we believe our work now meets the standard for NeurIPS and we hope the reviewers agree. In light of this, we humbly request the reviewers to consider raising their scores.

We find precisely *the same type of design choices* deliver strong empirical results (as in previous benchmarks). Concretely, there is a preference for: 1) better calibrated ensemble penalties (more frequently used in the Bayesian DL literature); 2) longer rollout horizons and; 3) higher penalty weights. Also note that *thanks to feedback from Reviewer mit8, we have chosen to rename the penalties* to dispel any confusion in our comparison, making it clearer we are comparing *uncertainty penalties*, not algorithms. We have renamed as follows:  MOPO -> Max Aleatoric; MOReL -> Max Pairwise Diff; LOMPO -> LL Var (Log-Likelihood Variance); M2AC -> LOO KL (Leave-One-Out KL Divergence). This change will be made throughout the paper and will be shown in the tables in our responses.

| Environment | Dataset type | MOPO (authors) | Optimized (Ours) | SOTA |
|---|---|---|---|---|
| walker | random | 13.6 | 21.7 | 37.3 (MOReL) |
| walker | mixed | 39.0 | 65.8* | 56.0 (COMBO) |
| walker | medium | 17.8 | 79.7* | 77.8 (MOReL) |
| walker | med-exp | 44.6 | 97.1* | 96.1 (COMBO) |

These results include SoTA performance for 3/4 environments, with a particularly large margin in the mixed setting. With these results, our average D4RL MuJoCo score is 65.2 compared to 64.4 from MOReL and 64.1 from COMBO, further confirming our analysis will be important to future work. We do not include any additional algorithmic innovations, so simply show that *alternative design choices can provide a significant boost for existing algorithms*.

Note again that our policy evaluation criteria is *likely* more rigorous than some previous offline methods, as we take the mean performance of the last ten iterations. It is not clear what many offline methods do, but issues have been raised on open sourced code for instance [here](https://github.com/tianheyu927/mopo/issues/5)). We believe this indicates the design choice settings we found are more [reliable across training epochs](https://arxiv.org/abs/1912.05663), which is vital for selecting policy checkpoints under off-policy evaluation (i.e., when we cannot have access to the true environment).

Finally, we find for the first time that offline MBRL can be competitive in the Adroit domains compared with model-free methods, and indeed provide the best performance seen so far (e.g., CQL) on the hammer-cloned setting. We believe issues with the world model not accurately capturing the *sparse reward* may account for any major performance reduction, and that our work is an important step for future work towards bridging the gap between model-based and model-free methods for sparse reward tasks, especially in the offline setting where exploration is not possible. We will highlight this accordingly in the paper. We define MOPO to be the max performance with the default penalty, choosing $\lambda$, $h$ in {1,5}$^2$.

| Environment | Dataset type | MOPO (Ours) | Optimized (Ours) | CQL (Model-Free) |
|---|---|---|---|---|
| pen | cloned | 5.4 | 23.0 | 39.2 |
| pen | human | 6.2 | 19.0 | 37.5 |
| pen | expert | 15.1 | 50.6 | 107.0 |
| hammer | cloned | 0.2 | 5.2 | 2.1 |
| hammer | human | 0.2 | 0.5 | 4.4 |
| hammer | expert | 6.2 | 23.3 | 86.7 |

---

### Author Response · Authors · 2021-08-28
**Following up**

Dear reviewers,

We appreciate your feedback and comments, and indeed have made many changes to our paper and run new experiments as a result of carefully reading them. We would be very grateful if you could confirm whether our individual and general responses were able to address your concerns, and if you had any additional issues having read the other reviews.

Many thanks,

Revisiting Design Choices in Offline MBRL Authors

---

### Decision · Program_Chairs · 2021-09-27

**Decision:**

Reject

**Comment:**

Overall the reviews were thorough and pointed to a number of limitations of the original submission. The authors provided a series of responses, which included a significantly larger set of experimental results, adjustments to the paper text including fundamental adjustments to make the paper focus clear, and promises of additional paper adjustments. The reviewers and AC took all of this into account.

In the end, we are looking at a very different paper than what was originally submitted. It is likely a nice improvement, which is the intent of the review process. However, the delta between the original and new content is quite large and arguably points to the fact that a new submission is needed. Indeed, there are many new experiments that have been produced in a very compressed amount of time (this even includes new experimental domains). It seems most appropriate that these be used as the basis for an upcoming submission, where more time can be spent verifying and analyzing the results.